# Antimicrobial and Antiradical Activity of Extracts from Leaves of Various Cultivars of *Pyrus communis* and *Pyrus pyrifolia*

**DOI:** 10.3390/biom15060821

**Published:** 2025-06-05

**Authors:** Beata Żbikowska, Magdalena Kotowska, Andrzej Gamian, Katarzyna Patek, Katarzyna Matuła, Daria Augustyniak, Kamila Korzekwa, Zbigniew Sroka

**Affiliations:** 1Department of Pharmacognosy and Herbal Medicine, Wroclaw Medical University, Borowska 211a, 50-556 Wrocław, Poland; beata.zbikowska@umw.edu.pl; 2Laboratory of Molecular Biology of Microorganisms, Hirszfeld Institute of Immunology and Experimental Therapy, Polish Academy of Sciences, Rudolfa Weigla 12, 53-114 Wrocław, Poland; 3Department of Immunology of Infectious Diseases, Hirszfeld Institute of Immunology and Experimental Therapy, Polish Academy of Sciences, Rudolfa Weigla 12, 53-114 Wrocław, Poland; andrzej.gamian@hirszfeld.pl; 4Faculty of Pharmacy, Wroclaw Medical University, Borowska 211a, 50-556 Wrocław, Poland; 5Department of Pathogen Biology and Immunology, University of Wroclaw, S. Przybyszewskiego 63, 51-148 Wrocław, Poland; daria.augustyniak@uwr.edu.pl; 6Department of Microbiology, University of Wroclaw, S. Przybyszewskiego 63, 51-148 Wrocław, Poland; kamila.korzekwa@uwr.edu.pl

**Keywords:** pear leaf extracts, arbutin, hydroquinone, phenolic compounds, antibacterial activity, antioxidant activity

## Abstract

Certain plant raw materials are rich in antioxidant and antimicrobial compounds, which are highly valued in modern medicine. These include the leaves of various species and cultivars of pears. For our research, we chose the leaves of the common pear (*Pyrus communis*) and Asian pear (*Pyrus pyrifolia*). Four different extracts were obtained from all raw materials and were investigated for their antimicrobial and antioxidant activity. The content of total phenolics and flavonoids was measured using colorimetric methods, and antiradical activity was measured using DPPH and ABTS radical probes. The antimicrobial activity of extracts was measured using the disc diffusion method, and the amount of major antimicrobial components (hydroquinone and arbutin) was measured using the HPLC method. The highest amount of general phenols and flavonoids was found in ethyl acetate extracts in all cultivars, and the lowest amount of phenols was found in the remaining aqueous solution. The amount of general phenols positively correlated with the antiradical activity of extracts. The strongest antimicrobial activity against Gram-positive and Gram-negative pathogens corresponded to the highest content of hydroquinone and arbutin in ethyl acetate extracts. Extracts obtained from pear leaves showed an average content of phenolic compounds and average antiradical activity compared to extracts from other raw materials, especially green tea or bergenia leaves. The amount of hydroquinone was moderate, lower than that of arbutin. The antimicrobial activity of the extracts was moderate due to the average amount of hydroquinone, which is the main antimicrobial compound.

## 1. Introduction

The golden era of antibiotics was between 1940 and 1965, when they were discovered and introduced into modern medicine [1]. The improper usage of antibiotics has led to growing bacterial resistance, which presents an increasing threat to contemporary medicine. This increases the risk of life-threatening infections in hospital wards and facilitates, the spread of resistant bacteria from hospital units to the wider society, leading to an escalating prevalence of antibiotic-resistant pathogens [2,3,4,5,6,7,8,9,10,11,12]. The other reasons for bacterial resistance are the inappropriate use of antibiotics in the population in general, as well as the use of these drugs as feed additives in the nutrition of farm animals.

Particularly dangerous are infections with highly and multidrug-resistant Gram-negative bacteria such as *Pseudomonas aeruginosa*, *Klebsiella pneumoniae*, and *Acinetobacter baumannii*, which belong to the so-called ESCAPE group of pathogens (*Enterococcus faecium*, *Staphylococcus aureus*, *Klebsiella pneumoniae*, *Acinetobacter baumannii*, *Pseudomonas aeruginosa*, and *Enterobacter* spp.), causing most nosocomial infections [1].

The problem of antibiotic resistance is complicated by too few new effective antimicrobial drugs being developed, while antibiotics’ effectiveness is falling sharply [13]. In addition, economic problems will increase due to the rising costs of treatment, health care, and mortality. This significantly undermines the effectiveness of current antimicrobial therapies, raising serious concerns for the future fight against bacterial and fungal diseases in both developed and developing countries.

To alleviate these urgent problems, funding should be given to research laboratories engaged in the search for new antimicrobial drugs, either by chemical synthesis or by the search for naturally occurring antimicrobial compounds [14,15] or by looking for inhibitors of enzymes that break down antibiotics, e.g., carbapenemases [16]. A potential alternative or complement to antibiotics may be the use of bacteriophages, i.e., viruses that can kill bacteria [17,18,19], or phage-derived enzymes, including endolysins and depolymerases [20]. Phytotherapy is also of great interest based on the search for naturally occurring antimicrobial compounds in medicinal and other plants [14,15]. There are many such plants whose extracts are effective against clinically relevant pathogens. For example, the antimicrobial activities of cranberry extract inhibiting the growth and limiting the biofilm formation of *Enterococcus faecalis* isolated from urinary tract infections have been confirmed [21]. Likewise, extracts from *Syzygium aromaticum*, *Punica granatum*, and *Thymus vulgaris* were effective against bacteria such as *Bacillus cereus*, *Staphylococcus aureus*, *Escherichia coli*, *Pseudomonas aeruginosa,* and *Salmonella typhi,* which cause food poisoning diseases [22]. Finally, extracts from *Pistacia lentiscus*, *Brassica oleracea*, *Glycyrrhiza glabra*, *Camellia sinensis*, *Cinnamomum cassia, Allium sativum*, *Nigella sativa,* and the leaves of various species of the genus *Bergenia* and natural compounds such as phenols—resveratrol, curcumin, quercetin, arbutin, and hydroquinone—and the organosulfur compound allicin demonstrated high antimicrobial activity against *H. pylori* [23,24]. Many similar examples could be provided, but we will stop here, as this is not a review work.

In addition to antibiotic resistance, another very important task for modern medicine is combating free radicals and oxidative stress resulting from an imbalance between the generation and neutralization of reactive oxygen and nitrogen species (ROS, RNS), as their excess occurs during various metabolic disorders and chronic diseases [25,26,27,28,29]. Excess of free radicals, ROS, and RNS leads to damage of important cellular structures such as lipids (the major components of membranes), causing the peroxidation of polyunsaturated fatty acids and oxidation of active and structural proteins, as well as DNA or RNA [30,31,32]. This, in turn, may lead to tissue damage, initiating many pathological processes, such as metabolic disorders (type II diabetes mellitus, obesity) [33], cancers [34], and chronic inflammatory disorders such as autoimmune diseases [35], which can also additionally generate an increase in ROS levels [36,37,38]. ROS affect wound healing, cell proliferation, granulation formation, and extracellular matrix formation and also harm wound healing [39]. Phytochemicals are increasingly used to treat inflammation-associated diseases by reducing the production of pro-inflammatory cytokines or regulating molecular mechanisms that synergize with the increased production of anti-inflammatory cytokines [40]. It has been documented that plant extracts from flax can inhibit human skin cell inflammation and cause remodeling of the extracellular matrix and wound closure activation [41]. The intensification of processes that generate free radicals destroys the body and accelerates the aging of organisms [42]. Plants and plant-derived compounds are therefore cited as a promising alternative or supportive components in strategies used to reduce microbial load or excessive oxidative stress [43]. The major groups of antimicrobial and antioxidant compounds made by plants comprise phenolics and polyphenols, terpenoids, alkaloids, and flavonoids. Among these compounds, phenols and polyphenols have the greatest power to reduce oxidative processes, and their activity strongly depends on their chemical structure [44]. Furthermore, mixtures of plant bioactive compounds are more effective than purified ones due to combination interactions [45].

The *Pyrus* species has been used in traditional medicine for over 2000 years. The genus *Pyrus* includes many species, such as Chinese pear (*Pyrus ussuriensis*; *Pyrus bretschneideri*), Asian pear (*Pyrus pyrifolia*), and the common pear (*Pyrus communis*) [46]. Different parts of pears, including leaves, contain both antimicrobial (arbutin and hydroquinone) [47,48] and antioxidant (polyphenols) compounds [24].

In this work, extracts prepared from leaves of different cultivars of *Pyrus pyrifolia* and *Pyrus communis* were tested for antimicrobial and antioxidant activity. The general amount of phenols was determined with colorimetric methods, and the amounts of hydroquinone and arbutin were measured with high-pressure liquid chromatography (HPLC).

## 2. Materials and Methods

### 2.1. Raw Material

The leaves used as research raw material were obtained from an experimental orchard of Wrocław University of Environmental and Life Sciences. There were three cultivars of Asian pear (*Pyrus pyrifolia* (Burm.f.) Nak.—cultivars *Chojuro*, *Kosui*, *and Shu-Li*) and three cultivars of common pear (*Pyrus communis* L.—cultivars *Dicolor*, *Clapp’s favourite*, and *Radana*). Voucher specimens were deposited in the Department of Horticulture (Wrocław University of Environmental and Life Sciences, Poland).

### 2.2. Reagents

The following reagents were used: analysis-grade methanol, Chempur Poland; gradient-grade methanol, Merck, Darmstadt, Germany; analytical-grade ethyl acetate, Chempur, Poland; Folin–Ciocalteu’s reagent, Fluka, Switzerland, analytical-grade sodium carbonate, Chempur, Poland; DPPH (2,2-diphenyl-1-picrylhydrazyl) free radical, Sigma-Aldrich, Steinheim, Germany; ABTS (2,2ʹ-azino-bis(3-ethylbenzothiazoline-6-sulfonic acid) diammonium salt) and potassium persulfate Sigma-Aldrich Steinheim, Germany; analytical-grade anhydrous aluminum chloride, Chempur, Poland; arbutin and hydroquinone, Sigma-Aldrich, Steinheim, Germany.

### 2.3. Apparatus

Spectrophotometer Cecil 3021, Cambridge, UK;

Spectrophotometer Hitachi U5100, Tokyo, Japan;

The DIONEX UHPLC Ultimate 3000 apparatus (Thermo Fisher Scientific, Waltham, MA, USA).

### 2.4. Preparation of Extracts

Extracts were prepared according to the method described by Sroka et al. [24], with modifications. A sample of 50 g of dried and crushed leaves from various cultivars of *Pyrus pyrifolia* and *Pyrus communis* was extracted with 900 mL of 60% solution of methanol in water at 45 °C for 48 h. Then, the extract was filtered with filter paper (Filtrak 388). A total of 180 mL of the extract was condensed under reduced pressure and lyophilized to obtain **EA** extract. The remaining part of the solution, with a volume of 720 mL, was condensed under reduced pressure to dryness and dissolved in 600 mL of water at 45 °C. The obtained solution was stored in a refrigerator at 4 °C for two days. The precipitate formed was separated using a filter (Filtrak 388) and freeze-dried to obtain residue **EB**. After **EB** separation, the aqueous solution was exhaustively extracted with ethyl acetate using a separatory funnel. The ethyl acetate solution was condensed under reduced pressure to obtain residue **EC**. After ethyl acetate extraction, the remaining aqueous solution was condensed under reduced pressure and then freeze-dried to obtain residue **ED**. Extracts obtained from different cultivars of *Pyrus communis* were labeled by adding the last letter corresponding to the cultivar. Extracts obtained from the *Radana* cultivar were identified as **EAR**, **EBR**, **ECR**, and **EDR** (weights of extracts [g] 2.943, 0.831, 1.894, and 8.851 g); extracts from *Clapp’s favourite* as **EAF**, **EBF**, **ECF**, and **EDF** (3.573, 0.729, 1.53, and 8.088 g); and extracts from *Dicolor* as **EAD**, **EBD**, **ECD**, and **EDD** (2.858, 0.517, 2.286, and 8.803 g). Extracts from the *Pyrus pyrifolia Chojuro* cultivar were identified as **EAC**, **EBC**, **ECC**, and **EDC** (4.067, 0.126, 0.966, and 7.007 g); extracts from *Kosui* as **EAK**, **EBK**, **ECK**, and **EDK** (5.812, 0.369, 1.736, and 9.518 g); and extracts from *Shu li* as **EAS**, **EBS**, **ECS**, and **EDS** (5.278, 0.271, 2.072, and 6.166 g).

### 2.5. Colorimetric Measurement of Total Phenol Content

The total amount of phenolic compounds in extracts was measured using the colorimetric method outlined by Singleton and Rossi [49] with Folin–Ciocalteu phenol reagent [50,51], which is a mixture of phosphotungstate and phosphomolybdate.

To the test tubes, 7 mL of water and 0.5 mL of methanol solution of extract at 1 mg/mL for **EA**, **EB**, and **EC** and 2 mg/mL for extract **ED** were added, followed by 0.5 mL of Folin–Ciocalteu phenol reagent. After three minutes, 2 mL of a 20% solution of Na_2_CO_3_ in water was added to the samples. The samples were heated for 1 min in a boiling water bath. After cooling, the samples were centrifuged at 5500 RCF. The absorbance of supernatants was measured at a wavelength of 685 nm. Measurements were repeated five times. The amount of phenolic compounds was expressed in mg of phenols per mg of extract (gallic acid equivalents per mg of extract, GAE/mg) and per g of raw material (GAE/g).

### 2.6. Measurement of Flavonoids in Extracts

Flavonoid content in extracts was measured according to the method first proposed by Christ and Muller [52] using the aluminum chloride (AlCl_3_) method. Flavonoids form with aluminum chloride (III) complexes with a yellow color, which absorb at a wavelength λ = 430 nm.

A total of 1 mL of methanolic solution of extracts at 1 mg/mL for extracts **EA**, **EB**, and **EC** and 2 mg/mL for extract **ED** was added to the test tube, and then 1 mL of 2% aluminum chloride solution in methanol was added. After 30 min of storage of samples in a dark place, absorbance was measured at a wavelength λ = 430 nm in glass cuvettes with an optical path length of 1 cm. Measurements for each sample were repeated five times.

The amount of flavonoids was expressed in mg of flavonoids per mg of extracts (as quercetin equivalents per mg of extracts, QE/mg) and per g of raw material (QE/g).

### 2.7. Investigation of Antioxidant Activity of Extracts

#### 2.7.1. Investigation of Antioxidant Activity of Extracts Using ABTS^•+^˙ Radical

The reduction in the cationic radical ABTS^•+^ determines the activity [53], which is obtained by the oxidation of ABTS (2,2′-azino-bis(3-ethylbenzothiazoline-6-sulfonic acid) diammonium salt) with potassium persulfate (K_2_S_2_O_8_). The color of the cationic radical is green-blue, the intensity of which decreases in the presence of the reductants, such as phenolic compounds. This is partly achieved through HAT (hydrogen atom transfer) and ET (electron transfer) types of reaction. The ABTS radical solution was prepared as follows. The solution ABTS in water at 7 mM was mixed with 2.45 mM aqueous solution of K_2_S_2_O_8_ in a volume ratio of 1:1, then left in a dark place for 16 h for the cation radical ABTS^•+^ to form. The final solution was diluted with 50% methanol solution so that the absorbance was equal to 1.

To measure antiradical activity, the extract was dissolved in a 1:1 methanol–water mixture (**EC** extract) or water **EA**, **EB**, and **ED** extracts. A reagent control sample was prepared by adding 20 µL of 50% methanol in water to 2 mL of ABTS^•+^ radical solution. The absorbance was measured at the beginning and after 1 min of the reaction at 25 °C using glass cuvettes with an optical path length equal to 1 cm. The maximal error was calculated using the total differential method.

The antiradical activity was calculated using the method described by Sroka et al. [54].

The antiradical activity was presented as the number of antiradical units per mg of extract *TAU_ABTS/mg_* and g of raw material *TAU_ABTS/g_*. One unit of antiradical activity is the amount of extract that causes the elimination of 1 mmole of ABTS^•+^ radical within 1 min, at the temperature of 25 °C, measured at wavelength λ = 734 nm.

#### 2.7.2. Calculation of the Number of *TAU_ABTS/mg_* and *TAU_ABTS/g_* Units

To calculate the number of antiradical units, the molar absorption coefficient ε for ABTS^•+^ equal to 1.5 × 10^4^ dm^3^ mol^−1^ cm^−1^ was used.

The number of antiradical units per mg of extracts was calculated according to the following equation:$$TAU_{ABTS/mg}=6.8\times 10^{-2}\frac{\left(A_{s0}-A_{s1})-(A_{c0}-A_{c1}\right)}{m}$$ where *TAU_ABTS/mg_* is the number of antiradical units per mg of extract; *A_s_*_0_ is the absorbance of the sample at the beginning of the reaction; *A_s_*_1_is the absorbance of the sample after 1 min of the reaction; *A_c_*_0_ is the absorbance of the control sample at the beginning of the reaction; *A_c_*_1_ is the absorbance of the control sample after 1 min of the reaction; *m* is the amount of extract [mg] in 1 mL of the sample.

Because *A_c_*_0_ − *A_c_*_1_ is always equal to 0, the equation was simplified to$$TAU_{ABTS/mg}=6.8\times 10^{-2}\frac{A_{s0}-A_{s1}}{m}$$

The amount of antiradical units per g of raw material was calculated according to the following equation:

$$TAU_{ABTS/g}=\frac{\left(TAU_{\frac{ABTS}{mgEA}}\cdot m_{EA})+(TAU_{\frac{ABTS}{mgEB}}\cdot m_{EB})+(TAU_{\frac{ABTS}{mgEC}}\cdot m_{EC})+(TAU_{\frac{ABTS}{mgED}}\cdot m_{ED}\right)}{W_{R}}$$ where *TAU_ABTS/g_* is number of antiradical units per g of raw material; *TAU_ABTS/mgEA_* is the number of antiradical units per mg of extract **EA**; *m_EA_* is total mass of extract **EA** [mg]; *TAU_ABTS/mgEB_* is the number of antiradical units per mg of extract **EB**; *m_EB_* is total mass of extract **EB** [mg]; *TAU_ABTS/mgEC_* is the number of antiradical units per mg of extract **EC**; *m_EC_* is total mass of **EC** extract [mg]; *TAU_ABTS/mgED_* is the number of antiradical units per mg of extract **ED**; *m_ED_* is total mass of extract **ED** [mg]; *W_R_* is total weight of raw material used for extraction [g].

#### 2.7.3. Measurement of Antiradical Activity of Extracts Using DPPH Radical

Extracts’ antiradical features were measured using the DPPH (2,2-diphenyl-picrylhydrazyl) radical, whose blue color changes in the presence of compounds with reducing properties such as phenols [55]. This is a HAT type of reaction involving the transfer of hydrogen atoms. DPPH in the radical form absorbs at λ = 515 nm; in the presence of antioxidants, absorption decreases.

A total of 2 mg of DPPH was dissolved in 54 mL of methanol. The solution was left until the absorbance stabilized at 515 nm. The absorbency of the DPPH solution was adjusted with methanol to a value of 1. The test was performed by adding 50 μL of extract solution to 2 mL of DPPH solution in methanol. The control sample was prepared by adding 50 μL of methanol, instead of extract solution, to DPPH. The absorbency was measured at the beginning of the reaction and after 1 min. Measurements were repeated 5 times. The maximal error was calculated using the total differential method.

The antiradical activity was presented as the number of antiradical units per mg of extract *TAU_DPPH/mg_* and g of raw material *TAU_DPPH/g_*. One unit of antiradical activity is the amount of extract that causes the elimination of 1 μmole of DPPH radical within 1 min, at the temperature of 25 °C, measured at wavelength λ = 515 nm.

#### 2.7.4. Calculation of the Number of *TAU_DPPH/mg_* Units

To calculate the number of antiradical units per mg of extract, the molar absorption coefficient ε for DPPH equal to 1.25 × 10^4^ dm^3^ mol^−1^ cm^−1^ was used.

The amount of antiradical units per mg of extracts was calculated according to the following equation:$$TAU_{DPPH/mg}=8.0\times 10^{-2}\frac{\left(A_{D0}-A_{D1})-(A_{C0}-A_{C1}\right)}{m}$$ where *TAU_515_*_/mg_ is the number of antioxidant units calculated per mg of extract; *A_D_*_0_ is the absorbance DPPH solution at the beginning of the reaction; *A_D_*_1_ is the absorbance of DPPH solution after 1 min of reaction; *A_C_*_0_ is the absorbance of the control sample at the beginning of the reaction; *A_C_*_1_ is the control sample after 1 min of reaction; *m* is the amount of extract in mg in 1 cm^3^ of sample.

Because *A_C_*_0_ − *A_C_*_1_ is always equal to 0, the equation was simplified to$$TAU_{DPPH/mg}=8.0\times 10^{-2}\frac{A_{D0}-A_{D1}}{m}$$

The amount of antiradical units per g of raw material was calculated according to the following equation:$$TAU_{DPPH/g}=\frac{\left(TAU_{\frac{DPPH}{mgEA}}\cdot m_{EA})+(TAU_{\frac{DPPH}{mgEB}}\cdot m_{EB})+(TAU_{\frac{DPPH}{mgEC}}\cdot m_{EC})+(TAU_{\frac{DPPH}{mgED}}\cdot m_{ED}\right)}{W_{R}}$$ where *TAU_DPPH/g_* is the number of antiradical units per g of raw material; *TAU_DPPH/mgEA_* is the number of antiradical units per mg of extract **EA**; *m_EA_* is the total mass of extract **EA** [mg]; *TAU_DPPH/mgEB_* is the number of antiradical units for mg of extract **EB**; *m_EB_* is the total mass of extract **EB** [mg]; *TAU_DPPH/mgEC_* is the number of antiradical units per mg of extract **EC**; *m_EC_* is the total mass of **EC** extract [mg]; *TAU_DPPH/mgED_* is the number of antiradical units per mg of extract **ED**; *m_ED_* is the total mass of extract **ED** [mg]; *W_R_* is the total weight of raw material used for extraction [g].

### 2.8. Measurement of Arbutin and Hydroquinone in Extracts Obtained from Leaves of Pyrus communis and Pyrus pyrifolia

Analysis of the amounts of arbutin and hydroquinone in extracts from leaves of *Pyrus communis* and *Pyrus pyrifolia* was performed with the Dionex UHPLC Ultimate 3000 apparatus (Thermo Fisher Scientific, USA) equipped with degasser (UltiMate 3000), pump (LPG-3400 SD), autosampler (WPS 3000) with dynamic mixing chamber (Reodyne 7770-362), temperature-controlled (10 °C) diode array detector (DAD-3000) and column thermostat (TCC 3000), Hypersil GOLD C18 column (250 × 4.6 mm, 5 µm), and C18 precolumn (10 × 4.6 mm, 5 µm), both from Thermo Scientific, USA. The analysis was carried out using a gradient as described by Fecka et al. [56]: A (water: formic acid, 98.5: 1.5, *v*/*v*) and B) acetonitrile: formic acid, 98.5: 1.5, *v*/*v*). The following gradient was applied: 0–30 min (3–30%, B in A), 30–33 min (30–70%, B in A), 33–45 min. (70%, B in A isocratic), and 45–48 min (70–100%, B in A). Then, the system was equilibrated to the starting condition of 3% B in A. The flow rate was 1.2 mL/min, the injection volume was 5 μL, and the temperature of analysis was 22 ± 1 °C. Spectral measurements were made at the wavelength of 190–800 nm, in steps of 1 nm and bunch width of 1 nm. Chromatograms were recorded at wavelengths 254, 271, 280, 283, 289, and 350 nm.

### 2.9. Investigation of Antimicrobial Activity of Extracts

#### 2.9.1. Bacterial Strains and Storage

The following Gram-positive and Gram-negative bacteria were used: (I) reference strains including *S. aureus* ATCC 25923, *E. faecalis* ATCC 25212, *E. coli* ATCC 25922, *P. aeruginosa* ATCC 27853, *H. pylori* ATCC 43504, and nontypeable *Haemophilus influenzae* (NTHi) ATCC 49247; (II) clinical strains including *S. aureus* MRSA 112, *E. faecalis* HLAR 78, *E. coli* ESBL 79, *P. aeruginosa* 256, nontypeable *Haemophilus influenzae* (NTHi 6), and *Bacillus subtilis* 21.

Most of the tested strains belong to the collection of the Department of Microbiology, University of Wroclaw. *H. influenzae* strains and *P. aeruginosa 256* belong to the Department of Pathogen Biology and Immunology, University of Wroclaw. *H. pylori* and *E. faecalis* were kindly lent from the Department of Microbiology, Wroclaw Medical University. Strains were stored frozen at −70 °C in relevant media, as described below, supplemented with 20% glycerol.

#### 2.9.2. Cultivation of Microorganisms

For experiments, the aerobic/facultative anaerobic bacterial strains (*S. aureus*, *E. faecalis*, *E.coli*, *P. aeruginosa*, and *B. subtilis*) were refreshed on Trypticase Soy Agar (TSA) (Becton Dickinson and Company, Cockeysville, MD, USA) at 37 °C for 18 h, whereas the microaerophilic strains were cultivated on Mueller–Hinton with 5% horse blood and NAD (*H. influenzae*, 24 h) or Columbia Agar with 5% sheep blood plus (*H. pylori*, 48 h) (Thermo Scientific, Wesel, Germany). *H. influenzae* and *H. pylori* were incubated in a humidified atmosphere with 5% CO_2_ and in a gas pack (GENbag microaer) with atmosphere of 80% N_2_, 10% H_2_, and 10% CO_2_ (bioMerieux, Craponne, France), respectively.

#### 2.9.3. Measurement of Antimicrobial Activity of Extracts

Plant extracts were diluted to a concentration of 100 mg/mL in DMSO, mixed vigorously on a shaker until completely dissolved, portioned, and stored in a refrigerator until use. For microbiological purposes, the disc diffusion method was used as recommended by EUCAST (Disc Diffusion Method for Antimicrobial Susceptibility Testing—Version 9.0, January 2021). Aerobic/facultative anaerobic bacteria and *H. influenzae* were inoculated at a concentration of ~1.5 × 10^8^ cfu/mL, referring to McFarland, 0.5 in 0.085% NaCl on Mueller–Hinton Agar 2. *H. pylori* was inoculated at a concentration of ~12 × 10^8^ cfu/mL, corresponding to McFarland 4, in Schaedler broth + vit. K3 (bioMerieux, Marcy-l’Etoile, France). Ten microliters with 1 mg of each extract were applied to 6 mm diameter blotting paper discs placed on Mueller–Hinton Agar 2 for aerobic/facultative anaerobic strains and Mueller–Hinton agar supplemented with 5% horse blood and NAD for microaerophilic strains. The plates were incubated for 18–24 h at 37 °C, except for those with *H. pylori*, which were incubated for 48 h. After incubation, growth inhibition zones were measured (as diameter given in millimeters). Analyses were performed in duplicate in three independent experiments. The positive control was a relevant synthetic antibiotic. The negative control was DMSO alone.

#### 2.9.4. Investigation of Inhibition of Growth of Extracts on *Campylobacter Jejuni* NCTC 11168

Due to increasingly frequent infections with Gram-negative *Campylobacter jejuni*, especially in the elderly and immunocompromised [57], the effect of this bacterium was additionally examined as described below.

The *Campylobacter jejuni* strain NCTC 11168 was obtained from the collection of the Laboratory of Molecular Biology of Microorganisms, Hirszfeld Institute of Immunology and Experimental Therapy, Polish Academy of Sciences.

*C. jejuni* was cultured in Brain Heart Infusion (Oxoid) (BHI) to OD_600_ = ~0.6 and then diluted to OD_600_ = 0.1 in BHI. Next, diluted bacteria culture was spread evenly with a cotton swab on the surface of square plates (12 cm × 12 cm) containing 40 mL of Columbia blood agar base medium supplemented with 10% fetal bovine serum and an antibiotic mix according to Contreras et al. [58].

Plant extracts were dissolved in DMSO to a concentration of 100 mg/mL and applied on Whatman GF/F sterile discs with a diameter of 6 mm (10 µL/disc). Discs were then placed on the surface of the inoculated plates. H_2_O_2_ (5 µL of 2% solution/disc) was used as a positive control.

*C. jejuni* was cultivated for 48 h at 42 °C under microaerobic conditions (3.5% H_2_, 6% O_2_, 7% CO_2_, and 85% N_2_) generated using the jar evacuation–replacement method (the Anaerobic Gas System PetriSphere).

Tests were perfromed in triplicate. Results are shown as mean ± standard deviation.

## 3. Results

### 3.1. Amounts of Phenolic Compounds and Flavonoids in Extracts and Raw Materials

The amounts of phenolic compounds in extracts and raw materials are presented in Table 1 and Table 2 and Figure 1, Figure 2 and Figure 3. The highest total phenol amount was measured in extract **ECK** (0.416 ± 0.011, GAE/mg) obtained from *P. pyrifolia* cultivar *Kosui* leaves. The highest amounts of phenolic compounds were generally detected in ethyl acetate extracts in all *P. pyrifolia* and *P. communis* cultivars with the following GAE/mg values: 0.401 ± 0.0015, 0.377 ± 0.0019, 0.347 ± 0.0008, 0.338 ± 0.009, and 0.334 ± 0.009 for extracts **ECF**, **ECD**, **ECR**, **ECC**, and **ECS**, respectively.

The lowest amount of total phenols was found for lyophilized aqueous residues: GAE/mg values 0.108 ± 0.003, 0.092 ± 0.002, 0.092 ± 0.0018, 0.083 ± 0.0004, 0.077 ± 0.0004, and 0.067 ± 0.002, for extracts **EDS**, **EDK**, **EDF**, **EDD**, **EDR**, and **EDC**, respectively.

The raw material with the highest amount of total phenols was the leaves of *P. pyrifolia Kosui* (52.37 ± 1.39, GAE/g). The smallest amount of phenolic compounds was found in the leaves of *P. pyrifolia Chojuro* (28.88 ± 0.77, GAE/g) (see Table 2, Figure 3).

The amount of flavonoids was much lower than the total phenols. As with the total phenol content, the highest amount of flavonoids was detected in the ethyl acetate extracts **ECD**, **ECF**, **ECR**, **ECC**, **ECK**, and **ECS**, and the lowest amount of flavonoids was measured for the lyophilized water residues **EDD**, **EDF**, **EDR**, **EDC**, **EDK**, and **EDS**. The highest content of flavonoids was found in the leaves of *P. pyrifolia Shu Li* (7.39 ± 0.21 QE/g), and the lowest in the leaves of *P. pyrifolia Chojuro* (4.31 ± 0.12 QE/g) (see Table 1 and Table 2, Figure 1, Figure 2 and Figure 3).

### 3.2. Antiradical Potential of Extracts and Leaves of P. pyrifolia and P. communis

The antiradical features of extracts and leaves are presented in Table 1 and Table 2 and Figure 1, Figure 2 and Figure 4.

Antiradical activity of extracts was measured using special probes, namely, cation radical ABTS^•+^ (2,2′- azino-bis(3-ethylbenzothiazoline-6-sulfonic acid) diammonium salt) and radical DPPH (2,2-diphenyl-1-picrylhydrazyl). When ABTS was used, the activity was presented as the number of antiradical units per mg of extract and grams of raw materials (*TAU_ABTS/mg_*, *TAU_ABTS/g_*). When the DPPH radical was used, the antiradical activity was presented as *TAU_DPPH/mg_* and *TAU_DPPH/g_*.

As with the content of phenolic compounds, the greatest antiradical activity was detected for the ethyl acetate extracts in both the DPPH and ABTS tests.

Namely, for ABTS tests for extracts **ECR**, **ECD**, **ECF**, **ECK**, **ECS**, and **ECC,** the activity was, respectively, 2.93 ± 0.19, 2.91 ± 0.17, 2.81 ± 0.15, 2.47 ± 0.095, 2.278 ± 0.115, and 2.277 ± 0.055 *TAU_ABTS/mg_*. For DPPH tests, the activity was, respectively, 1.12 ± 0.03, 1.24 ± 0.04, 1.02 ± 0.04, 1.661 ± 0.047, 1.334 ± 0.04, and 1.257 ± 0.052 *TAU_DPPH/mg_*. The lowest antiradical activity levels were observed for the aqueous residues labeled as **EDD**, **EDF**, **EDR**, **EDC**, **EDK**, and **EDS**. Pearson’s correlation coefficient (r) between the amount of total phenols and antiradical activity was 0.85 for the ABTS test and 0.82 for the DPPH test.

When we calculated the antiradical potential per g of raw material for the ABTS test, the highest number of antiradical units was observed for leaves from *P. communis Clapps favoritae* cultivar (501.6 ± 35.0 *TAU_ABTS/g_*), and leaves from *P. communis Dicolor* (450.3 ± 35.2 *TAU_ABTS/g_*). The lowest activity was noted for leaves from *P. pyrifolia Chojuro* (294.7 ± 29.6 *TAU_ABTS/g_*). In the DPPH tests, the highest number of antiradical units was observed for *P. pyrifolia Kosui* (244.86 ± 12.42 *TAU_DPPH/g_*), and the lowest was observed for *P. pyrifolia Chojuro* (127.47 ± 8.95 *TAU_DPPH/g_*).

### 3.3. Antibacterial Activity of Extracts Obtained from Leaves of Different P. pyrifolia and P. communis Cultivars

During this study, eleven types of extracts were tested in the *Pyrus pyrifolia* group, and twelve extracts were tested in the *Pyrus communis* group. Analyzing the antibacterial activity of ethyl acetate extracts **ECD**, **ECF**, and **ECR** of *Pyrus communis* (Table 3) and **ECC**, **ECK**, **ECS**, and water residue **EDS** of *Pyrus pyrifolia* (Table 4), a wide spectrum of activity against both aerobic/facultative anaerobic and microaerophilic strains belonging to Gram-positive and Gram-negative pathogens was recorded. All bacterial reference strains from the American Type Culture Collection (ATCC), except *E. faecalis* ATCC 25212, were sensitive to most of these specified extracts, although the level of sensitivity was different. Most of these extracts possess activity against clinical multidrug-resistant *P. aeruginosa* 256 and nontypeable *H. influenzae* NTHi6.

The most susceptible to the plant extract tested, as documented by the largest inhibition zones, were *H. pylori* ATCC 43504 and, to a lesser degree, *B. subtilis* 21. For both strains, growth inhibition was observed for most of the extract formulations derived from *Pyrus pyrifolia* and for *H. pylori* ATCC 43504, and also in the case of the extract from *Pyrus communis.* The above-mentioned most potent extracts from both *Pyrus* species also act against *S. aureus* ATCC 25 923 and *Haemophilus influenzae* (reference and clinical strains).

As documented, the ethyl acetate extracts containing the most hydroquinone (Table 5 and Table 6) showed the strongest antibacterial potency (Table 3 and Table 4), suggesting that hydroquinone is responsible for the observed antimicrobial effect. This is consistent with our previous research documenting that extracts rich in hydroquinone usually exhibited high antimicrobial activity, where the dependence of activity on the concentration of hydroquinone is proportional [24,59,60].

The research described in this study showed (Table 3 and Table 4) that the strongest inhibition of bacterial growth was observed for *Helicobacter pylori*, both for extracts obtained from leaves of *Pyrus communis* and *Pyrus pyrifolia*, where the greatest activity was observed for ethyl acetate extracts (**ECD**, **ECF**, **ECR**, **ECC**, **ECK**, and **ECS**). In the case of *H. pylori*, the growth inhibition zone diameter for the **ECS** extract was 11 ± 0.8 mm. In addition to ethyl acetate extracts, *Helicobacter pylori* growth was strongly inhibited by **EBK** extract (11 ± 1.7 mm). The aqueous residues (**EDD**, **EDF**, **EDR**, **EDC**, and **EDK**) did not show any activity against any strain, except for the **EDS** extract (Table 6), which contained a significant amount of hydroquinone. Some antibacterial activity was observed for *Bacillus subtilis*, *Haemophilus influenzae*, and *Staphylococcus aureus* ATCC 25923, especially for extracts from *Pyrus pyrifolia* leaves.

Low antimicrobial activity was observed against *Escherichia coli* (reference and clinical strains) and *Pseudomonas aeruginosa* (reference and clinical strains). No antimicrobial activity was observed against the strains *Staphylococcus aureus* MRSA and *Enterococcus faecalis* (reference and clinical strains).

The sum of diameters of inhibition zones for all strains (*SZD*) was highest for ethyl acetate extracts in the following decreasing order: **ECC** (84.2) > **ECK** (78.1) > **ECD** 72.4 > **ECF** (69.5) > **ECR** (69.4) > **ECS** (65.2).

In general, extracts from leaves of *Pyrus pyrifolia* were more active than extracts from leaves of *Pyrus communis*. The correlation coefficient *r* between the sum of inhibition zones (*SZD*) for different bacterial species and the hydroquinone content for *Pyrus communis* extracts was 0.9, while for *P. pyrifolia* extracts, it was 0.72. For arbutin, the corresponding correlation coefficient values were 0.68 and 0.59, respectively.

Although the correlation coefficient between arbutin content and antimicrobial effect is relatively high, arbutin is not an active substance, as our previous control research has shown. Our research demonstrated that the most active substance is hydroquinone [60].

Due to the increasing clinical significance of infections caused by *Campylobacter jejuni*, we additionally performed tests on the ability of extracts to inhibit the growth of this bacterium. These tests were performed independently in the Hirszfeld Institute of Immunology and Experimental Therapy laboratory. The results are graphically displayed in Figure 5.

The diameters of inhibition zones (mm) for extracts from both *P. communis* and *P. pyrifolia* were **EAD** (7.5 ± 1.2), **EBD** (7.2 ± 0.3), **ECD** (7.5 ± 0.2), **EDD** (no activity), **EAF** (7.4 ± 0.9), **EBF** (8.1 ± 0.5), **ECF** (9.5 ± 0.4), **EDF** (6.6 ± 0.6), **EAR** (7.3 ± 0.4), **EBR** (7.1 ± 0.7), **ECR** (7.4 ± 0.7), **EDR** (6.5 ± 0.5), **EAC** (8.3 ± 1.1), **EBC** (6.5 ± 0.5), **ECC** (11.2 ± 0.8), **EDC** (7.6 ± 0.4), **EAK** (9.4 ± 1.4), **EBK** (9.0 ± 1.6), **ECK** (13.5 ± 2.3), **EDK** (7.8 ± 1.1), **EAS** (9.8 ± 1.7), **EBS** (8.7 ± 2.2), **ECS** (13.2 ± 3.5), and **EDS** (15.8 ± 2.3).

*P. pyrifolia* extracts showed significantly greater activity than extracts from *P. communis* against *Campylobacter jejuni.* The most potent inhibitors of *C. jejuni* growth were **ECC**, **ECK**, **ECS,** and **EDS** extracts. The correlation coefficient between the activity of extracts against *C. jejuni* and the content of hydroquinone and arbutin was, respectively, *r* = 0.85, *r* = 0.24 (Figure 6). The high correlation between the hydroquinone content in the extracts and the inhibition of the growth of *C. jejuni* indicates that the main antimicrobial component is hydroquinone. Arbutin seems to be less important. Our previous research led to a similar conclusion [60].

### 3.4. Amounts of Arbutin and Hydroquinone in Extracts and Raw Materials

As described above, hydroquinone significantly inhibits the growth of different species of bacteria.

The amounts of hydroquinone and arbutin in extracts investigated in this research are demonstrated in Table 5 for *P. communis* and Table 6 for *P. pyrifolia* cultivars. A graphical presentation of the results is shown in Figure 7 for hydroquinone and Figure 8 for arbutin. Selected chromatograms are presented in Figure 9. The highest amount of arbutin was measured in **ECK** and **ECR** extracts from *P. pyrifolia Kosui* (148.2 ± 0.14) and *P. communis Radana* (145.9 ± 0.11) leaves, respectively. A lower value was obtained for leaf extract **ECC** of *P. pyrifolia Chojuro* (92.7 ± 0.014) and then, in decreasing order, **ECF**, **EAK**, **EAR**, **ECD**, **EDK**, **EAF**, **EAC**, **EDF**, **EAS**, **ECS**, **EDC**, **EBK**, **EDR**, **EAD**, **EBF**, **EBC**, **EDD**, **EBD**, **EDS**, **EBS**, and **EBR**.

The highest amount of hydroquinone was observed for extract **EDS** (8.08 ± 0.204), then **ECS** (5.71 ± 0.041), **ECF** (4.95 ± 0.020), and **ECC** (4.63 ± 0.014), and, in decreasing order, **ECK**, **ECR**, **ECD**, **EAK**, **EBF**, **EAS**, **EAD**, **EBD**, **EBC**, **EAC** = **EBK**, **EBS** = **EAR**, **EBR**, **EAF**, **EDC** = **EDD**, **EDR**, **EDK**, and **EDF**.

The contents of hydroquinone and arbutin were also calculated per g of raw material (Table 5 and Table 6). The highest amount of hydroquinone was found in the leaves of the *Shu li* cultivar (1.517 mg/g raw material), which also contained the least arbutin (9.503 mg/g). The lowest amount of hydroquinone was found in the leaves of *Chojuro* (0.461 mg/g), and the most arbutin (24.751 mg/g) was present in *Kosui* leaves.

## 4. Discussion

Since increasing bacterial resistance to antibiotics is a global threat, combating microbial resistance requires coordinated efforts that include the development of new conventional antibiotics and alternative antimicrobials. One of these is plant-derived compounds that can serve as phytotherapeutic agents used in good treatment alternatives.

We investigated the antimicrobial activity of a variety of plant extracts (leaves of *Bergenia crassifolia*, *Bergenia cordifolia*, and some species of the *Pyrus* genus), and we found high inhibition of the growth of dangerous and/or clinically relevant pathogens, including *Pseudomonas aeruginosa*, MRSA, *Helicobacter pylori*, *Enterococcus faecalis*, and *Escherichia coli* ESBL [24,60].

In our previous studies, we obtained highly active methanolic and ethyl acetate extracts from leaves of different cultivars of the pear (Conference and Shinseiki) containing arbutin, hydroquinone, and a large amount of phenols, mainly against Gram-positive strains such as *Staphylococcus aureus*, including MRSA strains, and *Bacillus subtilis* [24,60].

Some ethyl acetate extracts rich in hydroquinone showed high antibacterial activity against Gram-negative *Helicobacter pylori* that was similar to antibiotics such as erythromycin, chloramphenicol, and tetracycline [24,60].

In some cases, we obtained good effects of extracts containing a high content of hydroquinone against clinically relevant pathogens [60] such as Gram-positive *Enterococcus faecalis*, which may cause endocarditis, urinary tract infections, prostatitis, and intra-abdominal infection [61,62], and Gram-negative *Pseudomonas aeruginosa*, which may infect patients with carcinomas, immunodeficiency, cystic fibrosis, and chronic obstructive pulmonary disorder (COPD and serious infection requiring ventilation [63,64,65]. Antibacterial activity against Gram-negative *E. coli* ATCC 25922 and *E. coli* ESBL has also been demonstrated. It is worth adding that numerous *E. coli* strains can cause gastrointestinal tract infections [66], urinary tract infections [67], and Lesniowski–Crohn’s disease [68].

It can be found in the literature that hydroquinone and hydroquinone-rich extracts are inactive against Gram-negative bacteria [69]. Our study showed that extracts containing large amounts of hydroquinone are active against Gram-negative bacteria, but their activity is lower than that of Gram-positive bacteria [24].

In this study, the extracts showed moderate antimicrobial activity and an average content of hydroquinone, and these extracts contained a large amount of arbutin, which is microbiologically inactive. Thanks to the high content of arbutin (hydroquinone β D-glucopyranoside), the extracts have a high antimicrobial potential because arbutin relatively easily hydrolyzes (e.g., in alkaline conditions in urine), resulting in the release of active hydroquinone. Such hydrolysis usually leads to an increase in the antimicrobial activity of arbutin-rich extracts [70].

Further studies will be conducted to increase the antimicrobial potential of the extracts.

Another important problem for modern medicine is chronic and degenerative diseases [71], the severity of which may be related to oxidative and free radical processes that occur in the body, especially if the intensity of these processes is high [72].

Due to their chemical structure, phenolic compounds, abundant in plants, usually have strong antioxidant properties and inhibit free radical processes [73]. Their activity strongly depends on their chemical structure [54]. Flavonoids are an important group of plant phenols with significant antioxidant potential, which can weaken inflammatory processes by interfering with the arachidonic acid metabolism pathways [74].

In this study, we measured the total content of phenols and flavonoids in plant extracts and raw materials and evaluated their antiradical and antioxidant features.

Our research showed that the highest antiradical activity was found in ethyl acetate extracts with the highest content of phenolic compounds, while the weakest antioxidant activity and the lowest amount of phenols were found for water residues. The correlation coefficient (*r*) between the amount of total phenols and antiradical activity was 0.85 for ABTS tests and 0.82 for DPPH tests.

The leaves of the *Kosui* cultivar showed the strongest antiradical activity and contained the highest amount of phenolic compounds, while leaves from the *Chojuro* cultivar had the weakest antiradical properties and contained the least phenolic compounds.

In general, both phenolic content and antiradical properties of both extracts and raw materials are high, although weaker than in the case of extracts from bergenia leaves [60] and green and black tea leaves [75].

## 5. Conclusions

1. Extracts obtained from leaves of different *Pyrus communis* and *Pyrus pyrifolia* cultivars exhibited antioxidant and antiradical activity.

2. The antiradical activity of extracts was positively correlated with the amount of total phenolic compounds in them.

3. The highest antibacterial activity was observed for ethyl acetate extracts. 

4. Antibacterial activity positively correlated with the hydroquinone content and, to a lesser extent, with the amount of arbutin.

## Figures and Tables

**Figure 1 biomolecules-15-00821-f001:**
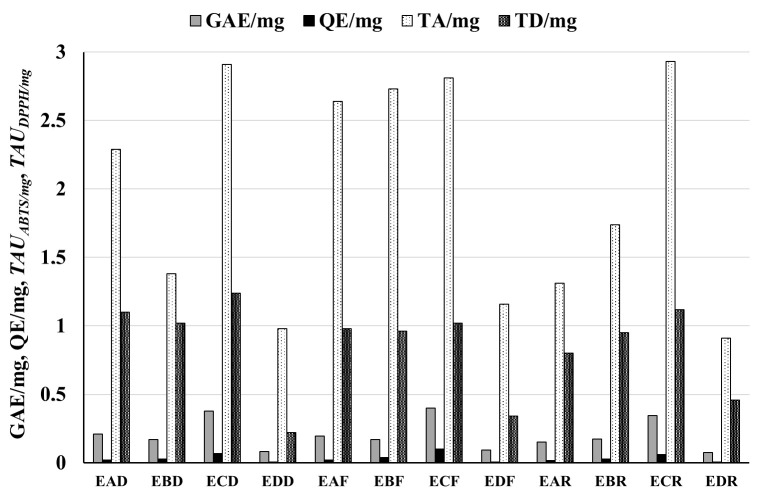
Total phenolic (GAE/mg) and flavonoid (QE/mg) content and antiradical activity (*TAU_ABTS/mg_* presented in the figure as TA/mg, *TAU_DPPH/mg_* as TD/mg) of extracts from different cultivars of *Pyrus communis*. Antiradical activity is expressed as the number of antiradical units per mg of extracts with the use of cation radical ABTS^•+^ and radical DPPH. An explanation of abbreviations of extracts is provided in the section on preparation of extracts.

**Figure 2 biomolecules-15-00821-f002:**
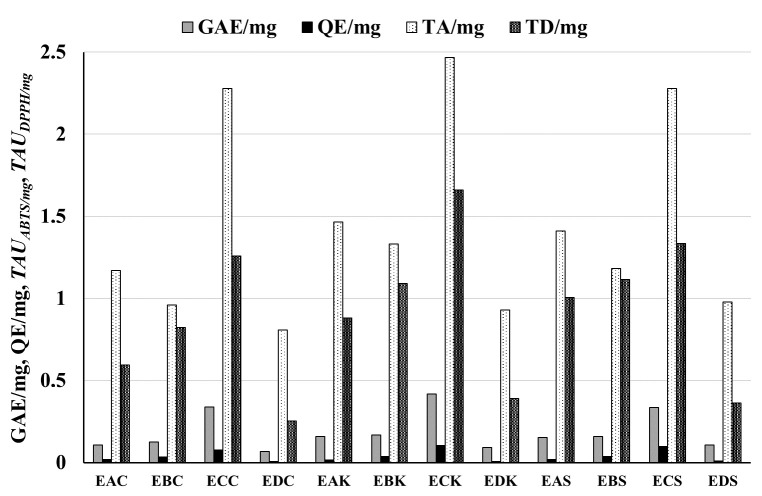
Total phenolic (GAE/mg) and flavonoid (QE/mg) content and antiradical activity (*TAU_ABTS/mg_*, presented in the figure as TA/mg, *TAU_DPPH/mg_* as TD/mg) of extracts from different cultivars of *Pyrus pyrifolia*. Antiradical activity is expressed as the number of antiradical units per mg of extracts with the use of cation radical ABTS^•+^ and radical DPPH. An explanation of abbreviations of extracts is provided in the section on preparation of extracts.

**Figure 3 biomolecules-15-00821-f003:**
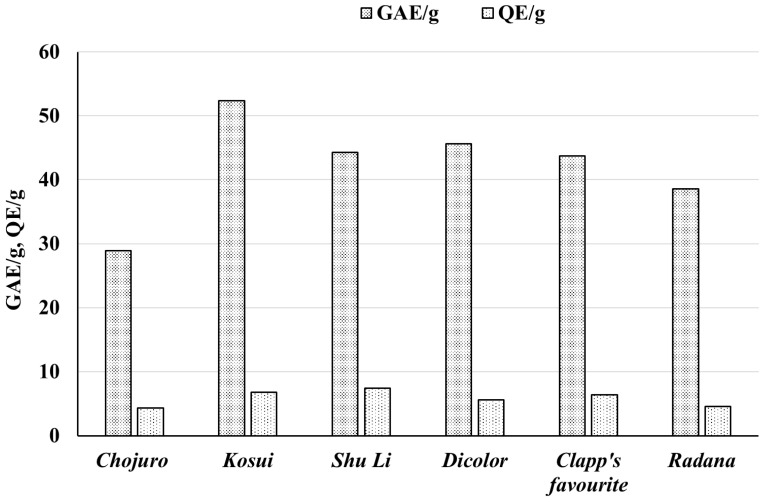
Total phenol (GAE/g) and flavonoid (QE/g) content in leaves of different cultivars of *Pyrus communis* and *Pyrus pyrifolia* expressed in mg per g of raw material. An explanation of abbreviations of extracts is provided in the section on preparation of extracts.

**Figure 4 biomolecules-15-00821-f004:**
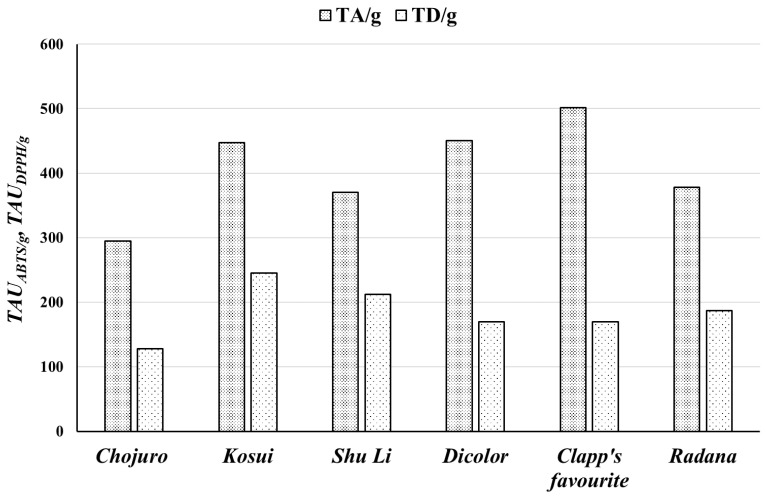
The number of antiradical units per g of raw material (*TAU_ABTS/g_ TAU_DPPH/g_*) measured using ABTS^•+^ cation radical and DPPH radical. An explanation of abbreviations of extracts is provided in the section on preparation of extracts.

**Figure 5 biomolecules-15-00821-f005:**
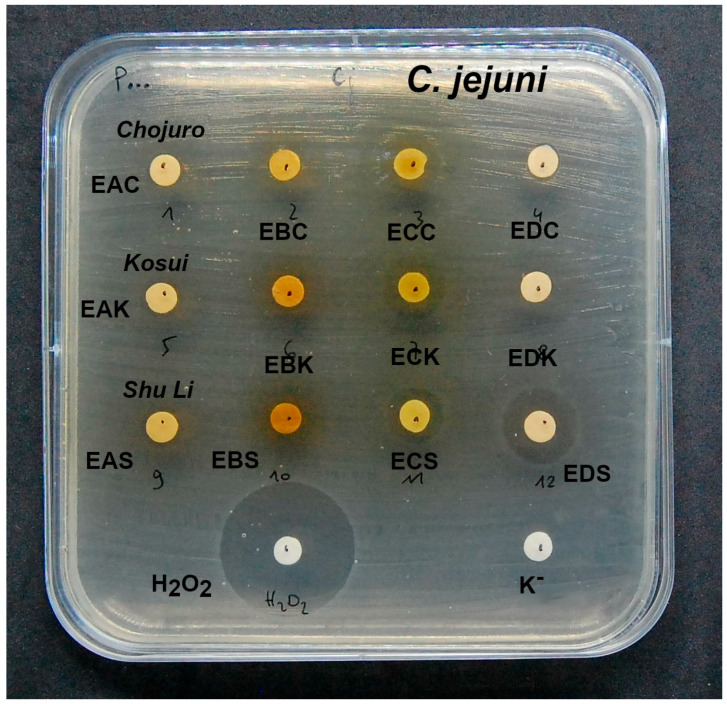
Figure showing the plate with the zones of inhibition of growth of *Campylobacter jejuni* in the presence of extracts from *Chojuro* cultivar, *Pyrus pyrifolia,* with the strongest antibacterial activity. An explanation of abbreviations of extracts is given in the section on preparation of extracts.

**Figure 6 biomolecules-15-00821-f006:**
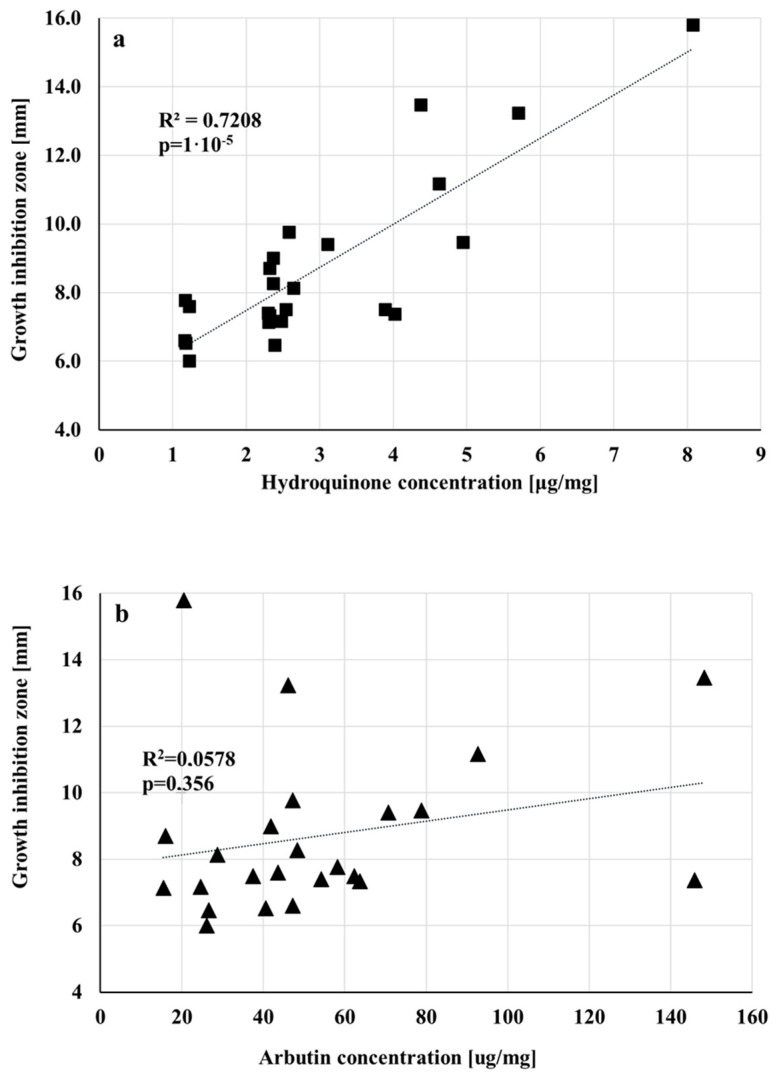
Graphical demonstration of the dependence of bacterial growth inhibition of *Campylobacter jejuni* on the concentration of hydroquinone (**a**) and arbutin (**b**) in extracts. The differences between samples are significant when *p* < 0.05. The dotted line indicates a trend line.

**Figure 7 biomolecules-15-00821-f007:**
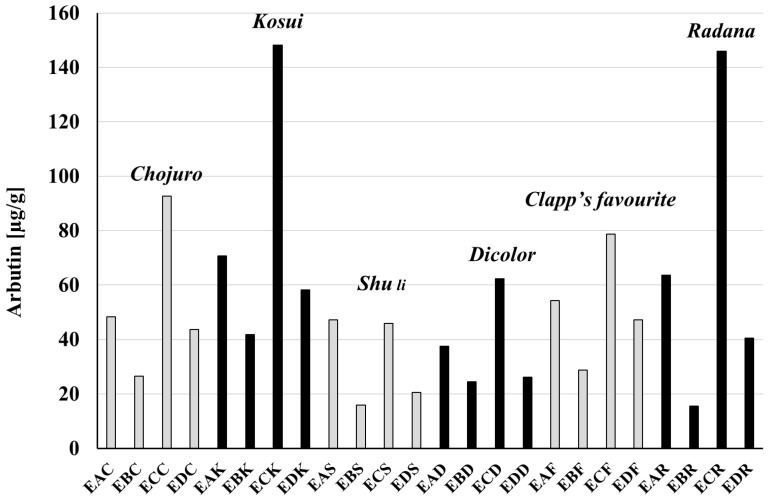
Comparison of arbutin content in extracts. An explanation of abbreviations of extracts is given in the section on preparation of extracts.

**Figure 8 biomolecules-15-00821-f008:**
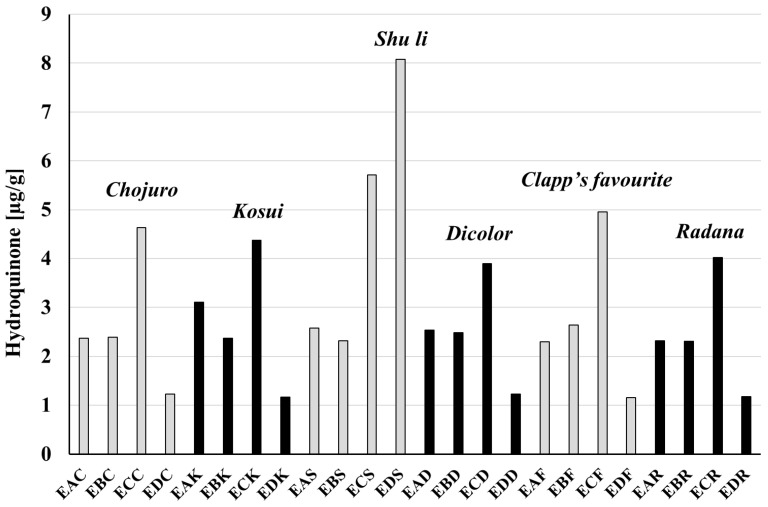
Comparison of hydroquinone content in extracts. An explanation of abbreviations of extracts is given in the section on preparation of extracts.

**Figure 9 biomolecules-15-00821-f009:**
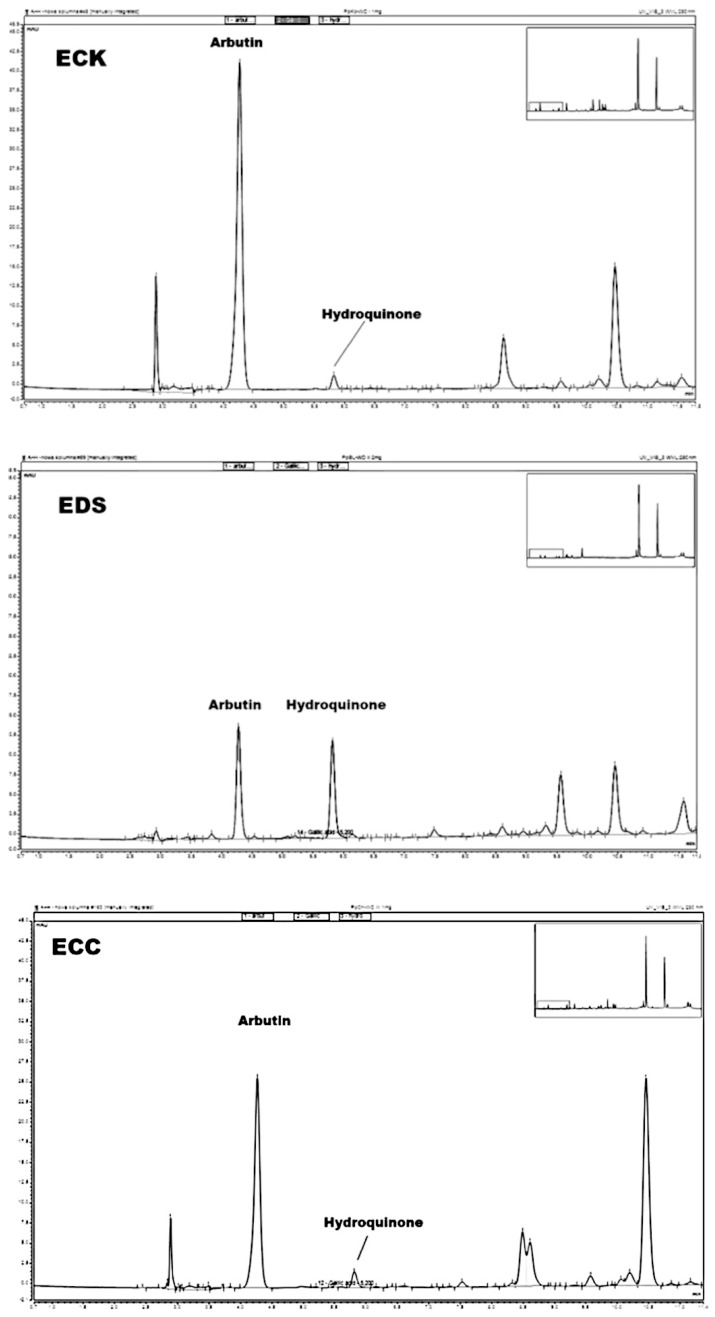
HPLC chromatograms of selected extracts with marked peaks of arbutin and hydroquinone. ECK—extract with the highest amount of arbutin, EDS—extract with the highest amount of hydroquinone, ECC—extract with the highest antimicrobial activity (*SZD* value).

**Table 1 biomolecules-15-00821-t001:** The amount of general phenols (GAE/mg, GAE/g) and flavonoids (QE/mg, QE/g) in mg of extracts and g of dry leaves of *Pyrus communis* L. Antiradical activity of extracts from leaves of *Pyrus communis* measured with the use of ABTS^•+^ and DPPH radicals and expressed as the number of antiradical units per mg of extract (*TAU_ABTS/mg_,*
*TAU_DPPH/mg_*) and grams of raw material (*TAU_ABTS/g_*, *TAU_DPPH/g_*) (±maximal error − ME).

Raw Material	Extracts	GAE/mg	GAE/g	QE/mg	QE/g	*TAU_ABTS/mg_*	*TAU_ABTS/g_*	*TAU_DPPH/mg_*	*TAU_DPPH/g_*
* **Pyrus communis “Dicolor”** *	**EAD**	0.211 ± 0.0010	45.61 ± 1.16	0.019 ± 0.0001	5.61 ± 0.08	2.29 ± 0.17	450.3 ± 35.2	1.10 ± 0.06	169.4 ± 13.9
	**EBD**	0.171 ± 0.0018		0.028 ± 0.0001		1.38 ± 0.07		1.02 ± 0.05	
	**ECD**	0.377 ± 0.0019		0.068 ± 0.0002		2.91 ± 0.17		1.24 ± 0.04	
	**EDD**	0.083 ± 0.0004		0.006 ± 0.0001		0.98 ± 0.04		0.22 ± 0.03	
* **Pyrus communis “Clapp’s favourite”** *	**EAF**	0.197 ± 0.0017	43.73 ± 2.07	0.022 ± 0.0001	6.42 ± 0.05	2.64 ± 0.14	501.6 ± 35.0	0.98 ± 0.05	169.9 ± 15.4
	**EBF**	0.170 ± 0.0025		0.039 ± 0.0001		2.73 ± 0.22		0.96 ± 0.05	
	**ECF**	0.401 ± 0.0015		0.100 ± 0.0002		2.81 ± 0.15		1.02 ± 0.04	
	**EDF**	0.092 ± 0.0018		0.007 ± 0.0001		1.16 ± 0.04		0.34 ± 0.04	
* **Pyrus communis** * * **“Radana”** *	**EAR**	0.152 ± 0.0010	38.58 ± 0.59	0.016 ± 0.0001	4.57 ± 0.10	1.31 ± 0.07	378.3 ± 27.9	0.80 ± 0.05	186.5 ± 11.0
	**EBR**	0.174 ± 0.0003		0.029 ± 0.0002		1.74 ± 0.09		0.95 ± 0.05	
	**ECR**	0.347 ± 0.0008		0.059 ± 0.0003		2.93 ± 0.19		1.12 ± 0.03	
	**EDR**	0.077 ± 0.0004		0.005 ± 0.0001		0.91 ± 0.04		0.46 ± 0.01	

**Table 2 biomolecules-15-00821-t002:** The amount of general phenols (GAE/mg, GAE/g) and flavonoids (QE/mg, QE/g) in mg of extracts and g of dry leaves of *Pyrus pyrifolia* L. Antiradical activity of extracts from leaves of *Pyrus pyrifolia* measured with the use of ABTS^•+^ and DPPH radicals and expressed as the number of antiradical units per mg of extract (*TAU_ABTS/mg_*, *TAU_DPPH/m_*_g_) and grams of raw material (*TAU_ABTS/g_*, *TAU_DPPH/_*_g_) (±ME).

Raw Material	Extracts	GAE/mg	GAE/g	QE/mg	QE/g	*TAU_ABTS/mg_*	*TAU_ABTS/g_*	*TAU_DPPH/mg_*	*TAU_DPPH/g_*
* **Pyrus pyrifolia** * * **“Chojuro”** *	**EAC**	0.108 ± 0.003	28.88 ± 0.77	0.017 ± 0.001	4.31 ± 0.12	1.17 ± 0.07	294.7 ± 29.6	0.595 ± 0.028	127.47 ± 8.95
	**EBC**	0.126 ± 0.003		0.035 ± 0.001		0.96 ± 0.070		0.821 ± 0.040	
	**ECC**	0.338 ± 0.009		0.075 ± 0.002		2.277 ± 0.055		1.257 ± 0.052	
	**EDC**	0.067 ± 0.002		0.006 ± 0.001		0.807 ± 0.089		0.254 ± 0.013	
* **Pyrus pyrifolia** * * **“Kosui”** *	**EAK**	0.159 ± 0.004	52.37 ± 1.39	0.016 ± 0.001	6.81 ± 0.19	1.466 ± 0.054	447.59 ± 26.79	0.881 ± 0.024	244.86 ± 12.42
	**EBK**	0.168 ± 0.005		0.036 ± 0.001		1.332 ± 0.072		1.092 ± 0.034	
	**ECK**	0.416 ± 0.011		0.103 ± 0.003		2.466 ± 0.095		1.661 ± 0.047	
	**EDK**	0.092 ± 0.002		0.006 ± 0.001		0.929 ± 0.039		0.389 ± 0.010	
* **Pyrus pyrifolia** * * **“Shu Li”** *	**EAS**	0.154 ± 0.004	44.31 ± 1.17	0.020 ± 0.001	7.39 ± 0.21	1.411 ± 0.058	369.93 ± 24.94	1.006 ± 0.064	212.22 ± 14.03
	**EBS**	0.159 ± 0.004		0.037 ± 0.001		1.181 ± 0.069		1.114 ± 0.034	
	**ECS**	0.334 ± 0.009		0.099 ± 0.003		2.278 ± 0.115		1.334 ± 0.040	
	**EDS**	0.108 ± 0.003		0.008 ± 0.001		0.977 ± 0.051		0.364 ± 0.0010	

**Table 3 biomolecules-15-00821-t003:** Inhibiting zones [mm] of bacterial growth for extracts from leaves of *Pyrus communis*.No antibacterial activity was found for extracts EDD, EDF, and EDR, and extract EBD was not investigated. S.a.—*Staphylococcus aureus*, *E.c.—Escherichia coli*, *P.a.—Pseudomonas aeruginosa*, *B.s.—Bacillus subtilis*, *NTHi*—nontypeable *Haemophilus influenza,* AMC—augmentin, C—chloramphenicol, CIP—ciprofloxacin, CPX—ciclopirox, DO—doxycillin, FOX—cefoxitin, IMP—imipenem, SXT—sulfamethoxazole, TZP—tazobactum, *SZD*—the sum of inhibition zone diameter for all bacterial strains for extract (±standard deviation—SD). 0 means no antibacterial activity.

Extract	*S.a.*ATCC 25923	*S.a*. MRSA	*E.c.*ATCC 25922	*E.c.*ESBL	*P.a.*ATCC 27853	*P.a.* 256	*B.s.*	*NTHi* ATCC 47249	*NTHi*6	*H.p.* ATCC 43504	*SZD*
**EAD**	0	0	0	0	0	0	0	0	0	9.2 ± 0.68	9.2 ± 0.68
**ECD**	11 ± 0.9	0	8.7 ± 0.5	0	9.3 ± 1	7.2 ± 0.6	8 ± 0.9	9.7 ± 0.6	8.5 ± 0.6	10 ± 0.52	72.4 ± 5.62
**EAF**	0	0	0	0	0	0	0	0	0	10.5 ± 1.2	10.5 ± 1.2
**EBF**	0	0	0	0	0	0	0	0	0	8.3 ± 0.52	8.3 ± 0.52
**ECF**	12.2 ± 0.4	0	7.1 ± 0.5	0	9 ± 0.7	6.9 ± 0.2	7.9 ± 1.4	9.2 ± 1	7.9 ± 0.9	9.3 ± 1.03	69.5 ± 6.13
**EAR**	0	0	0	0	0	0	0	0	0	8.7 ± 0.93	8.7 ± 0.93
**EBR**	10.3 ± 0.8	0	0	0	0	0	0	0	0	9.7 ± 1.03	20 ± 1.83
**ECR**	9.8 ± 1.6	0	7.6 ± 0.7	0	8.8 ± 0.8	8 ± 0.9	7.6 ± 0.5	8.7 ± 0.8	8.9 ± 0.9	10 ± 0.89	69.4 ± 7.09
**Antibiotic**	CPX (16)	CIP(25)	CIP(25)	AMC(16)		CIP(30)	AMC(23)		IMP(25)	DO(31)	
	FOX(30)		AMC(25)	SXT(30)		IMP(22)			AMC(23)	C(20)	
	CIP(27)		SXT(34)	TZP(21)		TZP(22)			SXT(28)		
**DMSO**	0	0	0	0	0	0	0	0	0	0	

**Table 4 biomolecules-15-00821-t004:** Inhibiting zones [mm] of bacterial growth for extracts from leaves of *Pyrus pyrifolia*. No antibacterial activity was found for extracts EBC, EDC, and EDK. S.a.—*Staphylococcus aureus*, *E.c.—Escherichia coli*, *P.a.—Pseudomonas aeruginosa*, *B.s.—Bacillus subtilis*, *N.T.H.i.*—nontypeable *Haemophilus influenza*, *H.p.—Helicobacter pylori. SZD*—the sum of inhibition zone diameter for all bacterial strains for extract [mm] (±SD). 0 means no antibacterial activity.

Extract	*S.a.*ATCC 25923	*S.a*. MRSA	*E.c.*ATCC 25922	*E.c.*ESBL	*P.a.*ATCC 27853	*P.a.* 256	*B.s.*	*N.T.H.i.* ATCC 47249	*N.T.H.i.*6	*H.p.* ATCC 43504	*SZD*
**EAC**	0	0	0	0	0	0	8.2 ± 0.6	0	0	8.4 ± 0.8	16.6 ± 1.4
**ECC**	13 ± 2	7.6 ± 0.5	9 ± 0.9	0	8.7 ± 1	8 ± 0.6	11.8 ± 1.5	8.9 ± 0.7	8.4 ± 0.9	8.8 ± 1.6	84.2 ± 9.7
**EAK**	0	0	0	0	0	0	9.8 ± 0.8	0	0	7.9 ± 0.8	17.7 ± 1.6
**EBK**	0	0	0	0	0	0	8.5 ± 1.2	0	0	11 ± 1.7	19.5 ± 2.9
**ECK**	14.4 ± 3	0	7.3 ± 0.9	0	7.4 ± 0.5	7.7 ± 0.8	12.1 ± 0.8	9.8 ± 0.8	9.4 ± 0.5	10.0 ± 1.6	78.1 ± 8.9
**EAS**	0	0	0	0	0	0	0	0	0	9.2 ± 0.29	9.2 ± 0.29
**EBS**	0	0	0	0	0	0	8.8 ± 1.2	0	0	10 ± 3.5	18.8 ± 4.7
**ECS**	11.2 ± 1	0	0	0	8.6 ± 1.1	7.8 ± 0.7	8.7 ± 0.7	9.1 ± 0.5	8.8 ± 0.8	11 ± 0.8	65.2 ± 5.6
**EDS**	11.4 ± 2	0	0	0	0	0	11.5 ± 1	9.3 ± 0.8	9.8 ± 0.3	7.6 ± 0.4	40.4 ± 4.5

**Table 5 biomolecules-15-00821-t005:** Amount of hydroquinone and arbutin in µg per mg of extract and in mg per g of raw material for *Pyrus communis* (± ME).

Cultivar	Extract	Hydrochinon [μg/mg Dry Weight]	Arbutin [μg/mg dr Weight]	Hydroquinone [mg/g Raw Material]	Arbutin [mg/g Raw Material]
* **Dicolor** *	**EAD**	2.54 ± 0.004	37.48 ± 0.141	0.565 ± 0.014	9.849 ± 0.213
	**EBD**	2.48 ± 0.007	24.56 ± 0.111		
	**ECD**	3.89 ± 0.033	62.34 ± 0.034		
	**EDD**	1.23 ± 0.005	26.14 ± 0.031		
* **Clapp’s** * * **favourite** *	**EAF**	2.30 ± 0.013	54.26 ± 0.076	0.542 ± 0.013	14.361 ± 0.314
	**EBF**	2.64 ± 0.026	28.71 ± 0.108		
	**ECF**	4.95 ± 0.020	78.79 ± 0.163		
	**EDF**	1.16 ± 0.004	47.27 ± 0.093		
* **Radana** *	**EAR**	2.32 ± 0.001	63.73 ± 0.204	0.536 ± 0.014	16.717 ± 0.369
	**EBR**	2.31 ± 0.012	15.49 ± 0.126		
	**ECR**	4.03 ± 0.045	145.91 ± 0.112		
	**EDR**	1.18 ± 0.009	40.57 ± 0.091		

**Table 6 biomolecules-15-00821-t006:** Amount of hydroquinone and arbutin in µg per mg of extract and in mg per g of raw material for *Pyrus pyrifolia* (±ME).

Cultivar	Extract	Hydrochinon [μg/mg Extract]	Arbutin [μg/mg Extract]	Hydroquinone [mg/g Raw Material]	Arbutin [mg/g Raw Material]
* **Chojuro** *	**EAC**	2.37 ± 0.005	48.30 ± 0.509	0.461 ± 0.012	11.899 ± 0.260
	**EBC**	2.39 ± 0.025	26.59 ± 0.085		
	**ECC**	4.63 ± 0.014	92.68 ± 0.161		
	**EDC**	1.23 ± 0.017	43.62 ± 0.014		
* **Kosui** *	**EAK**	3.11 ± 0.016	70.69 ± 0.188	0.753 ± 0.021	24.754 ± 0.533
	**EBK**	2.37 ± 0.018	41.85 ± 0.132		
	**ECK**	4.37 ± 0.040	148.24 ± 0.144		
	**EDK**	1.17 ± 0.009	58.21 ± 0.051		
* **Shu Li** *	**EAS**	2.59 ± 0.030	47.17 ± 0.150	1.519 ± 0.061	9.502 ± 0.216
	**EBS**	2.32 ± 0.010	15.94 ± 0.081		
	**ECS**	5.71 ± 0.041	46.05 ± 0.192		
	**EDS**	8.08 ± 0.204	20.50 ± 0.016		

## Data Availability

The original contributions presented in this study are included in the article. Further inquiries can be directed to the corresponding author.

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
