# Peer review of "Antimicrobial and Antiradical Activity of Extracts from Leaves of Various Cultivars of Pyrus communis and Pyrus pyrifolia"

_biomolecules, 2025, doi:10.3390/biom15060821_

Round 1

Reviewer 1 Report

Comments and Suggestions for Authors

The researchers prepared extracts from the leaves of the common pear (Pyrus communis) and Asian pear (Pyrus pyrifolia) and evaluated their antimicrobial and antioxidant activities. The highest amount of general phenols and flavonoids was found in ethyl acetate extracts in all cultivars. The antiradical activity of extracts positively correlated with the contents of general polyphenols. The most potent antimicrobial pathogen activity corresponded with the highest content of hydroquinone and arbutin (ethyl acetate extracts). On average, pear leaves had similar phenolic compounds and antiradical activity levels compared to extracts from other raw materials, such as green tea or Bergenia leaves.
This report thoroughly studies the antimicrobial and antioxidant activity of extracts from the leaves of various Pyrus communis and Pyrus pyrifolia cultivars and the possible associations with their polyphenol, hydroquinone and arbutin contents. However, since the evaluation of these factors in pear leaves has been an ongoing area of research by the group (refs [24][59,60]) and is also a major area of study by others, it is unclear what new or novel information is provided here. The antimicrobial effects are linked to the hydroquinone and arbutin contents, but this is just an association. The effects could as equally relate to other compounds, alone or in combination, in the extracts. In essence, this study just adds to the long list of ill-defined plant extracts that have potentially beneficial health effects. The authors must describe and emphasize what is new in the present study and how the findings add to our knowledge of the specific bioactive factors in pear leaves and their potential modes of action in vitro and in vivo. This information would help in the identification of compounds that could be taken forward for clinical study.
Ln 43-45 ‘This increases the risk of life-threatening infections in hospital wards and facilitates the spread of resistant bacteria from hospital units to the wider society, leading to the escalating prevalence of antibiotic-resistant pathogens ([2-12].’
This statement suggests that the misuse of antibiotics in hospitals is the leading cause of antibiotic resistance spread. While this may contribute to the overall problem, inappropriate use of antibiotics in the general population and as food additives or growth promoters for animals are the main causes.
Ln 142. Preparation of extracts. Is this a published method? Citations on the method or more details are necessary.
Ln 150 How was the precipitate dried?
Ln 371-372 & figures and tables. Were any statistical comparisons and analyses of the data carried out?
Ln 395 & 408 Figures 1 - 2. What are the y-axes?
Ln 496-497 ‘As documented in Tables 3 and Table 4, the ethyl acetate extracts containing the most hydroquinone showed the strongest antibacterial potency, suggesting that hydroquinone is responsible for the observed antimicrobial effect.' Where is the information about hydroquinone and, indeed, arbutin concentrations provided? This information should be mentioned here.
Ln 534-536 Is arbutin inactive or only lowly active compared to hydroquinone? Could the activity of arbutin depend on synergy with other components which may or may not be present in the extracts?
Discussion. Clarify throughout the text when the discussion refers to extracts, whether they are those in the present study, isolated in previous work or isolated by other researchers.
Antimicrobials. Are hydroquinone and arbutin the only antimicrobials in the extracts, or are others present in lower amounts? Do the antimicrobials in the extract act alone or in synergy?
Ln 628-630 ‘In this study, the extracts showed moderate antimicrobial activity and average hydroquinone content. Thanks to the high content of arbutin (hydroquinone β D-glucopy-ranoside), the extracts have a high antimicrobial potential.’
How does this conclusion fit with earlier statements that arbutin has little or no antimicrobial activity?

Author Response

Open Review

Quality of English Language

( ) The English could be improved to more clearly express the research.
(x) The English is fine and does not require any improvement.

Yes

Can be improved

Must be improved

Not applicable

Does the introduction provide sufficient background and include all relevant references?

(x)

( )

( )

( )

Is the research design appropriate?

(x)

( )

( )

( )

Are the methods adequately described?

(x)

( )

( )

( )

Are the results clearly presented?

(x)

( )

( )

( )

Are the conclusions supported by the results?

( )

(x)

( )

( )

Comments and Suggestions for Authors

This manuscript seems to be well designed and presents some relevant data regarding the antimicrobial activity and antiradical activity of the extracts from pears.

Which is the main novelty of this current manuscript and differences regarding the previous papers?

Our research intended to compare antimicrobial potential and the antioxidant activity of three different cultivars for each of the two main species of pears - Pyrus communis (a representative of the western = European pear group) and Pyrus pyrifolia (a representative of the oriental = Asian pear group) that we had not been previously studied. In previous papers, we analysed leaves of other cultivars collected at different times of the year.

How do the authors guarantee that the peaks of hydroquinone or arbutin correspond to these compounds, as the extracts contain many other compounds?

UHPLC-DAD chromatography was performed for hydroquinone and arbutin standards at different concentrations, qualitative analysis of the compounds was performed by comparing the retention times with arbutin and hydroquinone standards and comparing the spectra with standards in the range of 190-400 nm, which was possible thanks to the use of a DAD type detector. Quantitative analysis was performed after determining the peak areas of the identified compounds, comparing them with the appropriate arbutin and hydroquinone

Regarding the conclusions, it should not be in the form of topics.

Globally, the manuscript was too long, maybe some information in the introduction or in the methodology should be omitted or inserted as supplementary materials.

The conclusions presented are a summary of the most important observations resulting from the research.Such a system seems to me to be good.

Submission Date

22 April 2025

Date of this review

10 May 2025 17:52:07

Reviewer 2 Report

Comments and Suggestions for Authors

This manuscript seems to be well designed and presents some relevant data regarding the antimicrobial activity and antiradical activity of the extracts from pears.

Which is the main novelty of this current manuscript and differences regarding the previous papers?

How do the authors guarantee that the peaks of hydroquinone or arbutin correspond to these compounds, as the extracts contain many other compounds?

Regarding the conclusions, it should not be in the form of topics.

Globally, the manuscript was too long, maybe some information in the introduction or in the methodology should be omitted or be inserted as supplementary materials.

Author Response

Open Review 1

Quality of English Language

( ) The English could be improved to more clearly express the research.
(x) The English is fine and does not require any improvement.

Yes

Can be improved

Must be improved

Not applicable

Does the introduction provide sufficient background and include all relevant references?

(x)

( )

( )

( )

Is the research design appropriate?

( )

( )

( )

( )

Are the methods adequately described?

( )

(x)

( )

( )

Are the results clearly presented?

( )

(x)

( )

( )

Are the conclusions supported by the results?

( )

(x)

( )

( )

Comments and Suggestions for Authors

The researchers prepared extracts from the leaves of the common pear (Pyrus communis) and Asian pear (Pyrus pyrifolia) and evaluated their antimicrobial and antioxidant activities. The highest amount of general phenols and flavonoids was found in ethyl acetate extracts in all cultivars. The antiradical activity of extracts positively correlated with the contents of general polyphenols. The most potent antimicrobial pathogen activity corresponded with the highest content of hydroquinone and arbutin (ethyl acetate extracts). On average, pear leaves had similar phenolic compounds and antiradical activity levels compared to extracts from other raw materials, such as green tea or Bergenia leaves.

This report thoroughly studies the antimicrobial and antioxidant activity of extracts from the leaves of various Pyrus communis and Pyrus pyrifolia cultivars and the possible associations with their polyphenol, hydroquinone and arbutin contents. However, since the evaluation of these factors in pear leaves has been an ongoing area of research by the group (refs [24][59,60]) and is also a major area of study by others (see recent references), it is unclear what new or novel information is provided here. The antimicrobial effects are linked to the hydroquinone and arbutin contents, but this is just an association.

We studied various raw materials with high arbutin and hydroquinone content, such as pear leaves or leaves of Bergenia crassifolia and cordifolia, as well as leaves of other species of the Bergenia genus.

In addition, we examined for antimicrobial activity artificially created mixtures of phenolic compounds, hydroquinone, and arbutin. These data clearly indicate that the main active compound is hydroquinone [60].

The new elements in this work are previously unexplored for the antibacterial activity cultivars of P. communis (Dicolor, Clapp’s favourite, Radana) and P. pyrifolia (Chojuro, Kosui, Shu Li).

The effects could as equally relate to other compounds, alone or in combination, in the extracts. In essence, this study just adds to the long list of ill-defined plant extracts that have potentially beneficial health effects. The authors must describe and emphasize what is new in the present study and how the findings add to our knowledge of the specific bioactive factors in pear leaves and their potential modes of action in vitro and in vivo.

Extracts are a mixture of many substances that can affect the antimicrobial features of these extracts. Tannins, which are one of the groups of natural compounds with significant antimicrobial properties, are not extractable with 60% methanol, therefore, we assumed that tannins do not affect the antimicrobial activity of our extracts. Another active group with high antimicrobial properties is oils or resins, which occur in trace amounts in the raw materials tested. Therefore, above mentioned compounds are not important for antimicrobial activity of the studied extracts. As was mentioned previously, we prepared artificial mixtures of natural compounds in various combinations consisting of phenols such as tannins, phenolic acids, hydroquinone and arbutin [60]. Our research clearly indicated the strong antimicrobial properties of hydroquinone.

[60] Żbikowska. B., Franiczek, R., Sowa, A., Połukord, G., Krzyżanowska, B., Sroka, Z. Antimicrobial and antiradical activity of extracts obtained from leaves of five species of the genus Bergenia: identification of antimicrobial compounds. Microb. Drug Resist. 2017, 23, 771-780. https://doi.org/10.1089/mdr.2016.0251.

This information would help in the identification of compounds that could be taken forward for clinical study.

Ln 43-45 ‘This increases the risk of life-threatening infections in hospital wards and facilitates the spread of resistant bacteria from hospital units to the wider society, leading to the escalating prevalence of antibiotic-resistant pathogens ([2-12].’

This statement suggests that the misuse of antibiotics in hospitals is the leading cause of antibiotic resistance spread. While this may contribute to the overall problem, inappropriate use of antibiotics in the general population and as food additives or growth promoters for animals (GhimpeÈ›eanu et al. Antibiotic Use in Livestock and Residues in Food—A Public Health Threat: A Review. Foods, 11(10), 1430. https://doi.org/10.3390/foods11101430) are the main causes.

Inappropriate use of antibiotics in hospitals is one of the main causes of the increase in bacterial resistance to antibiotics. This is not the only cause; there are others, as in the opinion of the Reviewer, the inappropriate use of antibiotics in the population in general. Another serious cause of the increase in bacterial resistance to antibiotics is the use of these drugs as feed additives in the nutrition of farm animals.

Ln 142. Preparation of extracts. Is this a published method? Citations on the method or more details are necessary.

corrected according to Reviewer remarks

Ln 150 How was the precipitate dried?

The precipitates were separated from the aqueous solution using a filter and freeze-dried.

Ln 371-372 & figures and tables. Were any statistical comparisons and analyses of the data carried out?

The maximal error was calculated for three repetitions; statistical significance of the differences between samples was not determined.

Ln 395 & 408 Figures 1 - 2. What are the y-axes?

The figures have been corrected, the y-axis shows the values defined in the method sections: 2.5, 2.6, 2.7.1, 2.7.2, 2.7.3, and 2.7.4.

Ln 496-497 ‘As documented in Tables 3 and Table 4, the ethyl acetate extracts containing the most hydroquinone showed the strongest antibacterial potency, suggesting that hydroquinone is responsible for the observed antimicrobial effect.' Where is the information about hydroquinone and, indeed, arbutin concentrations provided? This information should be mentioned here.

The concentrations of arbutin and hydroquinone are given in Tables 5 and 6, determined on the basis of HPLC analysis. Text was corrected.

Ln 534-536 Is arbutin inactive or only lowly active compared to hydroquinone?

Yes, we have already shown in previous studies that hydroquinone is much more active than arbutin. Correlations are often high for hydroquinone and low for arbutin.

Could the activity of arbutin depend on synergy with other components which may or may not be present in the extracts?

I don’t think so, hydroquinone is main compound with antimicrobial activity, and is active independently of other compounds.

Discussion. Clarify throughout the text when the discussion refers to extracts, whether they are those in the present study, isolated in previous work or isolated by other researchers.

Antimicrobials. Are hydroquinone and arbutin the only antimicrobials in the extracts, or are others present in lower amounts? Do the antimicrobials in the extract act alone or in synergy?

Extracts are difficult to analyze due to their complex composition. We have made every effort  to determine the active substances to the maximum extent, synergy has little effect on the microbial activity of compounds

Ln 628-630 ‘In this study, the extracts showed moderate antimicrobial activity and average hydroquinone content. Thanks to the high content of arbutin (hydroquinone β D-glucopy-ranoside), the extracts have a high antimicrobial potential.’

How does this conclusion fit with earlier statements that arbutin has little or no antimicrobial activity?

Arbutin is a derivative of hydroquinone and is relatively easily hydrolyzed, e.g. in the conditions of alkaline urine pH, in vivo, thus increasing the concentration of active hydroquinone. We carried out such hydrolysis in our laboratory, obtaining a significant increase in the concentration of hydroquinone and the significant increase of antimicrobial activity of extracts. Text was corrected.

Recent references

Ribeiro et al. (2025). Phenolic Compounds from Pyrus communis Residues: Mechanisms of Antibacterial Action and Therapeutic Applications. Antibiotics, 14(3), 280. https://doi.org/10.3390/antibiotics14030280

Molinu et al. (2024). Mediterranean Wild Pear Fruits as a Neglected but Valuable Source of Phenolic Compounds. Resources, 13(6), 72. https://doi.org/10.3390/resources13060072

Jiao et al. (2024). Metabolomics Analysis of Phenolic Composition and Content in Five Pear Cultivars Leaves. Plants, 13(17), 2513. https://doi.org/10.3390/plants13172513

In our research [60], we did not find strong antimicrobial properties of simple phenols. Among phenolic compounds we tested, only hydroquinone showed significant antibacterial activity.

Submission Date

22 April 2025

Date of this review

05 May 2025 14:50:35

Reviewer 3 Report

Comments and Suggestions for Authors

In this study, the authors investigated antimicrobial and antioxidant activities of extracts from leaves of the common pear and Asian pear. The highest amount of general phenols and flavonoids was found in ethyl acetate extracts, and the lowest amount of phenols was found in the remaining aqueous solution. The strongest antimicrobial activity against gram-positive and gram-negative pathogens corresponded to the highest content of hydroquinone and arbutin in ethyl acetate extracts. In general, the experiments were well-performed.

1) Not all abbreviations have been introduced with full names, such DPPH, ABTS and HPLC in Abstract.

2) There are so many tables and figures without the titles and legends, such as the table in Page 4 and four figures in Pages 4-7.

3) There are so many abbreviations listed in the tables. Their full names should be introduced in the legends.

4) For all columns in the figures, error bars are missing. And no statistic difference was marked. And the units for Y-axis are missing.

5) Numbers in Figure 5 are too small to be seen clearly.

6) The P values of two regressions in Figure 7 should be shown. By the way, "R2=0,7208" should be R2=0.7208 and "R2=0,0578" should be R2=0.0578.

Comments on the Quality of English Language

1) Common pear (Pyrus communis) and Asian pear (Pyrus pyrifolia) are two species, but not two cultivars as mentioned in Abstract (Line 26).

2) At the end of Abstract, "The amount of hydroquinone was lower than that of arbutin, resulting in moderate antimicrobial activity (lower than in our previous study. Nevertheless, the extracts tested have a high antimicrobial potential due to the high arbutin content." I cannot understand the causality. The fact that hydroquinone was lower than arbutin does not necessarily mean a moderate antimicrobial activity. There are so many other chemicals with high antimicrobial activities. The last sentence means that all extracts tested have high antimicrobial potentials and high arbutin contents? By the way, where is the punctuation ")"?

3) Manuscript's title for some reference is missing, e.g. Ref. 19.

Author Response

Open Review 2

Quality of English Language

(x) The English could be improved to more clearly express the research.
( ) The English is fine and does not require any improvement.

Yes

Can be improved

Must be improved

Not applicable

Does the introduction provide sufficient background and include all relevant references?

( )

(x)

( )

( )

Is the research design appropriate?

( )

(x)

( )

( )

Are the methods adequately described?

( )

(x)

( )

( )

Are the results clearly presented?

( )

( )

(x)

( )

Are the conclusions supported by the results?

( )

(x)

( )

( )

Comments and Suggestions for Authors

In this study, the authors investigated antimicrobial and antioxidant activities of extracts from leaves of the common pear and Asian pear. The highest amount of general phenols and flavonoids was found in ethyl acetate extracts, and the lowest amount of phenols was found in the remaining aqueous solution. The strongest antimicrobial activity against gram-positive and gram-negative pathogens corresponded to the highest content of hydroquinone and arbutin in ethyl acetate extracts. In general, the experiments were well-performed.

1) Not all abbreviations have been introduced with full names, such DPPH, ABTS and HPLC in Abstract.

All abbreviations are explained in the “Materials and Methods” section when they appear for the first time in the main text, according to the guide for authors. We have not explained these abbreviations in the abstract so as not to extend the text too long, according to other reviewers.

2) There are so many tables and figures without the titles and legends, such as the table in Page 4 and four figures in Pages 4-7.

corrected

3) There are so many abbreviations listed in the tables. Their full names should be introduced in the legends.

All abbreviations are explained in the “Materials and Methods” section when they appear for the first time, according to the guide for authors.

4) For all columns in the figures, error bars are missing. And no statistic difference was marked. And the units for Y-axis are missing.

Errors (maximal error) are given in tables, adding error bars to figures reduces their clarity. The units for y-axis are included in the figures entered into the journal’s system.

5) Numbers in Figure 5 are too small to be seen clearly.

corrected

6) The P values of two regressions in Figure

corrected

7 should be shown. By the way, "R2=0,7208" should be R2=0.7208 and "R2=0,0578" should be R2=0.0578.

corrected

Comments on the Quality of English Language

1) Common pear (Pyrus communis) and Asian pear (Pyrus pyrifolia) are two species, but not two cultivars as mentioned in Abstract (Line 26).

Thank you for this comment, but I did not find a place in the text where a species was confused with a cultivar. Two species were studied, P. communis and P. pyrifolia, from each species three cultivars, well described in the raw material section.

2) At the end of Abstract, "The amount of hydroquinone was lower than that of arbutin, resulting in moderate antimicrobial activity (lower than in our previous study. Nevertheless, the extracts tested have a high antimicrobial potential due to the high arbutin content." I cannot understand the causality.

The text has been changed. The sentence about the significant antimicrobial potential of arbutin has been removed because it requires additional explanation.  This sentence with the explanation was placed in the “Discussion” section.

The fact that hydroquinone was lower than arbutin does not necessarily mean a moderate antimicrobial activity. There are so many other chemicals with high antimicrobial activities. The last sentence means that all extracts tested have high antimicrobial potentials and high arbutin contents? By the way, where is the punctuation ")"?

All inconsistencies were corrected in the new text

3) Manuscript's title for some reference is missing, e.g. Ref. 19.

corrected

Submission Date

22 April 2025

Date of this review

03 May 2025 04:14:39

Round 2

Reviewer 1 Report

Comments and Suggestions for Authors

The authors have addressed all issues raised in the review in a thorough manner. 

Reviewer 3 Report

Comments and Suggestions for Authors

I'm satisfied with the revision.